# Studies of the Formation of Inclusion Complexes Derivatives of Cinnamon Acid with α-Cyclodextrin in a Wide Range of Temperatures Using Conductometric Methods

**DOI:** 10.3390/molecules27144420

**Published:** 2022-07-10

**Authors:** Zdzisław Kinart, Renato Tomaš

**Affiliations:** 1Department of Physical Chemistry, Faculty of Chemistry, University of Lodz, Pomorska 163/165, 90-236 Lodz, Poland; 2Department of Physical Chemistry, University of Split, Ruđera Boškovića 35, HR-21000 Split, Croatia; renato.tomas@ktf-split.hr

**Keywords:** electric conductivities, α-cyclodextrin, aqueous solutions of sodium salts of phenolic acids, complex constants, thermodynamic function

## Abstract

The electrical conductivities of aqueous solutions of sodium salts of *trans*-4-hydroxycinnamic acid (*trans*-*p*-coumaric acid), *trans*-3,4-dihydroxycinnamic acid (*trans*-caffeic acid), *trans*-4-hydroxy-3-methoxycinnamic acid, (*trans*-ferulic acid) and *trans*-3-phenylacrylic acid (*trans*-cinnamic acid) with α-cyclodextrin were measured in the temperature range of 288.15 K–318.15 K. For the first time in the literature, using the limiting molar conductivity $\left(\Lambda_{m}^{o}\right)$ obtained from conductivity measurements, the values of the complexation constants (*K_f_*) of the salts of phenolic acid derivatives with α-cyclodextrin were determined using a modified low concentration chemical model (IcCM). An attempt was also made to analyze the individual thermodynamic functions $\Delta G^{o}$, $\Delta H^{o}$ and $\Delta S^{o}$ describing the complexation process as a function of temperature changes. The obtained results show that the process of formation of inclusion complexes is exothermic and is spontaneous.

## 1. Introduction

The main objectives of the presented study were to determine the values of the formation constants *K_f_* of inclusion complexes of cinnamonic acid and its selected derivatives: *trans*-4-hydroxycinnamic acid, *trans*-3,4-dihydroxycinnamic acid, *trans*-4-hydroxy-3-methoxycinnamic acid and *trans*-3-phenylacrylic acid with α-cyclodextrin based on conductometric measurements and analysis of the impact of these acids’ structure and temperature on the values of *K_f_.*

Cyclodextrins are cyclic oligosaccharides, consisting of six (α-CD), seven (β-CD), eight (γ-CD) or more (LR-CD) large-ring α-D-glucopyranose molecules linked with each other through an α-1,4-glycosidic bond, adopting a chair conformation. This contributes to the lack of free rotation of their rings around said bond and the formation of the CD molecule in the shape of a truncated cone (torus) [1,2]. Structural studies have shown that the hydroxyl groups that give the CD molecule a hydrophilic character assume their characteristic arrangement, giving it a three-dimensional structure. The -OH groups on the secondary carbon atoms, C2 and C3, are located on the wider edge of the ring, while the -OH groups on the primary carbon atom, C6, are located on the narrower one. The hydrogen atoms located in the C3 and C5 carbon atoms and the oxygen atom of the glucopyranose ring are directed toward the inside of the molecule, ensuring the hydrophobic nature of the torus [3,4].

Cyclodextrins are characterized by high solubility in water because they have free hydroxyl groups directed outside the molecule, and the macrocyclic gap in the cyclodextrin molecule is slightly hydrophobic [5,6]. At room temperature, γ-CD (23.2 g/100 mL of water) dissolves best, then α-CD (14.5 g/100 mL) and β-CD (1.85 g/100 mL of water). In addition, cyclodextrins do not show toxic properties, which has been confirmed by numerous studies carried out in many species of animals, including rats.

In nature, CDs are formed by bacterial digestion of polysaccharides (e.g., starch, amylose) and the product of this reaction is a mixture containing variable amounts of native CDs and linear and branched dextrins and proteins. This process involves cyclodextrin glucanotransferase, an extracellular bacterial enzyme produced by some strains of the genus Bacillus. It catalyzes the cleavage of polysaccharides, and then the assembly of linear products of their degradation into cyclic oligosaccharides in the intramolecular transglycosylation reaction [5,6].

Due to their characteristic spatial structure, CDs have a number of interesting properties. The water solubility of natural CDs is much lower than that of linear dextrins with an analogous number of glucopyranose rings. The reason for this may be the relatively high cohesive energy, which causes strong binding of CD molecules in the crystalline state, or the formation of intramolecular hydrogen bonds, which reduce the interaction of CD molecules with surrounding water molecules. The substitution of hydroxyl groups with other groups (e.g., hydroxypropyl) in CD molecules may increase their solubility [7]. An important and frequently used feature of CDs is their ability to form inclusion complexes with other chemical compounds, e.g., medicinal substances. Usually, these are complexes with 1:1 stoichiometry, where one CD molecule is combined with one substance molecule. The formation of such a complex takes place through the incorporation of a drug substance molecule into the CD gap [8,9]. As a result of this interaction, neither the formation of new covalent bonds nor the breaking of the existing covalent bonds occurs. The molecules of the drug substance undergo various conformational changes to maximize the use of van der Waals interactions, which play a key role in the formation of the drug substance–CD complex, both in the solution and in solid state. The formation of such complexes depends primarily on the structure and physicochemical properties of both substances. A necessary condition for their formation is the matching of the nonpolar drug substance group to the hydrophobic CD gap.

Phenolic acids have an antioxidant potential that is greatest for the cinnamic acid derivatives. They are widespread throughout the world of plants, and although they are characterized by a variety of chemical structures and pharmacological properties, they contain the same functional groups (hydroxyl and carboxyl groups) [10,11].

However, one can distinguish between simple benzoic acids, phenylacetic acids and cinnamon acids by the number of carbons in the side chain. The most common derivatives of cinnamic acid are its hydroxyl derivatives, i.e., caffeic acid, ferulic acid and *p*-coumaric acid.

Phenolic acids are commonly present in food and dietary supplements as bioactive ingredients.

Recent studies describing the activity of a structurally diverse range of phenolic acid salts against foodborne pathogens indicate that both phenolic acids and their salts can be used as natural chemical and food preservatives [12,13]. The subjects of such studies are most often *o*-, *m*- and *p*-coumaric, caffeic, and cinnamic acids [12,14,15]. One well-known sodium salt of the cinnamic acid derivative containing both hydroxyl and methoxy groups is sodium ferulate (SF).

This article presents precise measurements of the electrical conductivity of aqueous solutions of sodium salts of cinnamon acid and its selected derivatives: *trans*-4-hydroxycinnamic acid (*trans*-*p*-coumaric acid), *trans*-3,4-dihydroxycinnamic acid (*trans*-caffeic acid), *trans*-4-hydroxy-3-methoxycinnamic acid (*trans*-ferulic acid) and *trans*-3-phenylacrylic acid (*trans*-cinnamic acid). All conductivities were measured at temperatures (*T*) from 288.15 K to 318.15 K and in the concentration range (*c*) starting from ~1.6 ×10^−3^ to ~2.8×10^−3^ mol·dm^−3^. The tested salts were selected to show the influence of changes in their structure, i.e., the location and number of hydroxyl or methoxy groups, on their conductometric properties. The conductivity of the investigated complexes was assessed with the use of modified conductivity equations. For the first time, the values of the formation constants of inclusion complexes of the studied sodium salts and α-cyclodextrin were determined based on conductometric measurements. The thermodynamic functions $\Delta G^{o},$ $\Delta H^{o\;}$ and $\Delta S^{o}\;$ describing this process were also determined. On the basis of the obtained data, the inclusion processes that take place in the studied complexes were discussed.

## 2. Materials and Methods

### 2.1. Materials

High purity of *trans*-4-hydroxycinnamic acid (*trans-p*-coumaric acid), *trans*-3,4-dihydroxycinnamic acid (*trans*-caffeic acid), *trans*-4-hydroxy-3-methoxycinnamic acid, (*trans*-ferulic acid), *trans*-3-phenylacrylic acid (*trans*-cinnamonic acid) and α-cyclodextrin were used. All information regarding their purity and suppliers is presented in Table 1.

Double distilled, deionized and degassed water with a specific conductance better than 0.5 × 10^−6^ S∙cm^−1^ was used for the preparation of studied solutions.

### 2.2. Characterization Methods

All studied solutions were prepared using an analytical balance (Sartorius RC 210D) with an uncertainty of ±1 × 10^−5^ g. The exact experimental procedure for conductometric measurements is described elsewhere [16]. Conductivity measurements were taken on an RLC Wayne-Kerr 6430B conductivity meter with an uncertainty of 0.02% using a three-electrode probe similar to that described in the literature [17]. The chamber was calibrated with an aqueous solution of potassium chloride [18]. For all measurements, a BU 20F calibration thermostat (Lauda, Lauda-Königshofen, Germany) with stability better than 0.005 K was used and the temperature was checked using an Amarell 3000TH AD thermometer (Amarell, Kreuzwertheim, Germany). The thermostat was connected to a DLK 25 flow cooler (Lauda, Lauda-Königshofen, Germany).

Conductivity measurements were carried out at different frequencies: *v* = 0.2, 0.5, 1.0, 1.5, 2.0, 3.0, 5.0, 10.0 and 20.0 kHz. All measured conductivity values, *λ = 1/R_∞_*, were the results of extrapolation of cell resistance *R_∞_*(*ν*) to infinite frequency *R_∞_* = *lim**_ν_*_→*∞*_*R*(*ν*) using the empirical function *R*(*ν*) = *R_∞_* + *A*/*ν* (where parameter *A* is specific for the cell). Taking into account the sources of errors (calibration, sample purity, measurements), the estimated uncertainty of the measured conductivity values was estimated at ±0.05%.

The studied salts were obtained by mixing the appropriate amounts of acid and aqueous sodium hydroxide solution in a stoichiometric ratio of 1:1. The mixture was then heated and stirred to dissolve the acid and evaporate the solvent. More detailed procedures of the preparation of the salts of phenolic acids are described in the literature [12,16,19,20]. The salts obtained were washed with acetone and dried in an oven under reduced pressure in a Büchi glass oven B-580 at *T* = 373.15 K until constant weight. The synthesized salts were tested by ^1^H-NMR spectra to confirm the absence of impurities. The spectra of the synthesized salts were consistent with those reported in the literature [20,21,22]. The purity of the obtained salts was 99%. The solubility of the tested sodium salts in water in the studied concentration range was very good and thus did not pose any experimental problems in conductivity measurements.

## 3. Results

The molar conductivity (Λ*_m_*) for the tested compounds along with their molality (m) values are presented in Appendix A.

To convert molonity, $\tilde{m}$, (moles of electrolyte per kilogram of solution) into molarity, *c*, the values of the density gradients, *b*, have been determined independently and used in Equation (1a):(1a)$$c/\tilde{m}=\rho =\rho_{o}+b\cdot \tilde{m}$$
where *ρ_o_* is the density of the solvent and *ρ* is the density of the solution.

The measured densities of binary solutions of water and salts as a function of temperature are presented in Appendix A. Density gradients and molar conductivity Λ as a function of molality, *m* (moles of electrolyte per kilogram of solvent), and temperature are presented in Appendix A. The relationship among *m*, $\tilde{m}$ and *c* is as follows:(1b)$$\tilde{m}=c/\rho =m/(1+m\cdot M)$$
where *M* is the molar mass of the electrolyte.

In the studied solutions, we observed most probably the formation of inclusion complexes with the 1:1 stoichiometry between the cyclodextrin (CD) and the anion of the tested phenolic acids (Dod^−^):$$\mathrm{CD}+{\mathrm{Dod}}^{-}\;↔\;{\mathrm{DodCD}}^{-}$$

The formation constant (*K_f_*) of these complexes has the following form:(2)$$K_{f}=\frac{\left[{\mathrm{DodCD}}^{-}\right]}{[{\mathrm{Dod}}^{-}]\left[\mathrm{CD}\right]}$$

The obtained results of conductometric measurements make it possible to determine the values of these constants.

The molar conductivity of the sodium salt of a selected phenolic acid can be expressed by the molar conductivities of the acid salt with an uncomplexed anion (Λ*_NaDod_*) and the alkaline salt of the acid, where the anion forms an inclusion complex with cyclodextrin (Λ*_CDNaDod_*).

The molar conductivity of a solution is represented by the following relationship:(3)$$\Lambda =\frac{1000\cdot \;\kappa }{C_{NaDod}}$$
where *κ* is the specific conductivity in the unit (S∙m^−1^) and *C_NaDod_* is the molar concentration of sodium salts of phenolic acid.

The molar conductivity of the tested phenolic acid, calculated from Equation (3), can be expressed by the molar conductivities of the tested sodium salt of phenolic acid with an uncomplexed anion (Λ*_NaDod_*) and sodium salt of phenolic acid in which the anion forms an inclusion complex with cyclodextrin (Λ*_CDNaDod_*).

The molar conductivity of the tested solution is described by the following relationship:(4)$$\Lambda_{\mathrm{obs}}=\alpha \cdot \Lambda_{NaDod}+\left(1-\alpha \right)\cdot \Lambda_{CDNaDod}$$
where:(5)$$\alpha =\frac{\left[{\mathrm{Dod}}^{-}\right]}{\left[{\mathrm{Na}}^{+}\right]}$$
(6)$$1-\alpha =\frac{\left[{\mathrm{DodCD}}^{-}\right]}{\left[{\mathrm{Na}}^{+}\right]}$$
(7)$$C_{DodNa}=\left[{\mathrm{Na}}^{+}\right]=\left[{\mathrm{Dod}}^{-}\right]+\left[{\mathrm{DodCD}}^{-}\right]$$
(8)$$C_{\mathrm{CD}}=\left[\mathrm{CD}\right]+\left[{\mathrm{DodCD}}^{-}\right]$$

The constant of complex ion (DodCD^−^) formation is given by the following equation:(9)$$K_{f}=\frac{\left[{\mathrm{DodCD}}^{-}\right]}{\left[{\mathrm{Dod}}^{-}\right]\cdot \left[\mathrm{CD}\right]}$$

Taking into account Equations (7)–(9), we obtain the following:(10)$$K_{f}\cdot {\left[\mathrm{CD}\right]}^{2}+\left[K_{f}\cdot \left(C_{DodNa}-C_{\mathrm{CD}}\right)+1\right]\cdot \left[\mathrm{CD}\right]-C_{\mathrm{CD}}=0$$

By combining Equations (4)–(6) and (10), we get an equation that contains all the quantities from the experiment and two selectable quantities (*K_f_* and $\Lambda_{NaDodCD}$).

Taking this into account, the equation for the molar conductivity of the solution before adding the cyclodextrin takes the following form:(11)$$\Lambda =\left[K_{f}\cdot \left(c_{DodNa}-C_{\mathrm{CD}}\right)-1+\sqrt{K_{f}^{2}\cdot {\left(C_{\mathrm{CD}}-C_{DodNa}\right)}^{2}+2K_{f}\cdot \left(C_{DodNa}+C_{\mathrm{CD}}\right)+1}\right]\cdot \left[\frac{\Lambda_{NaDod}-\Lambda_{CDNaDod}}{2K_{f}\cdot C_{DodNa}}\right]+\Lambda_{NaDodCD}$$

The values of *K_f_* and conductivity Λ*_NaDodCD_* are selected by minimizing the sum:(12)$$\overset{n}{\underset{i=1}{\sum }}{\left(\Lambda_{exp}-\Lambda_{calc}\right)}^{2}$$
where *n* is the number of test solutions, Λ*_exp_* is the experimental molar conductivity calculated from Equation (3) and Λ*_calc_* is the molar conductivity calculated from Equation (11).

A review of the literature data [23,24] shows that the values of Λ*_NaDod_* and Λ*_NaDodCD_* may change and their values can be described by the following equations:(13)$$\Lambda_{NaDod}=\Lambda_{oNaDod}-S\cdot c_{NaDod}^{1/2}+E\cdot c_{NaDod}\cdot \ln c_{NaDod}+J_{1}\cdot c_{S}+J_{2}\cdot c_{NaDod}^{3/2}$$
(14)$$\Lambda_{NaDodCD}=\Lambda_{oCDNaDod}-S\cdot c_{NaDod}^{1/2}+E\cdot c_{NaDod}\cdot \ln c_{NaDod}+J_{1}\cdot c_{S}+J_{2}\cdot c_{NaDod}^{3/2}$$

The values of the parameters *S*, *E*, *J*_1_ and *J*_2_ are related to the relaxation and electrophoretic effects. However, it should be remembered that the values of *J*_1_ and *J*_2_ depend on the so-called parameter of maximum ion approximation.

## 4. Discussion

As can be seen in Appendix A, the molar conductivities for α-cyclodextrin with the acid ions in the given temperature range were also collected. On the basis of the analysis of the above values, it can be concluded that the increase in temperature causes an increase in the molar conductivity of the tested salts, which is most likely caused by a decrease in the viscosity of the solution and thus causes the greater mobility of ions. It can also be concluded that as the concentration of the ligand increases (i.e., by adding successive portions of α-cyclodextrin), the acid ions increase inclusion (causing a slow decrease in conductivity values). As can be seen, the molar conductivity of the studied anions decreases with increasing ligand concentration. This is surprising because according to the theories of conductivity, the analyzed values of molar conductivity should increase. Therefore, it should be assumed that the observed decrease in the conductivity value is due to the fact that some of the acid anions form complex ions with cyclodextrin, which are characterized by lower conductivity.

It should be emphasized that the values presented in Table 2, Table 3, Table 4 and Table 5 containing the data of the limiting conductivity and the complexation constant for the sodium salts of the phenolic acids under consideration perfectly correlate with each other, and we obtain linear values (see Figure 1).

The conductivity values decrease with the elongation of the number of carbon atoms in the structure, which confirms the correctness of the calculations made according to the theory of conductivity. It has also been found that salts containing one or two hydroxyl groups have similar values; hence, in this case, the increase in the number of hydroxyl groups in the molecule does not appear to have a significant effect on the values of molar limiting conductivity. As can be seen, the greatest changes in conductivity are observed in the case of the salt of cinnamic acid. We also observe here the highest value of limiting molar conductivity with increasing temperature. As the molar mass of the tested acid increases, these values begin to stabilize and increase in a similar manner. The values of theoretical conductivity increase when the temperature and molar mass of the studied anions increase, which is compatible with the theory of conductivity (see Figure 1).

However, these values are significantly lower than the conductivity values of the salt when the anion is not complexed. This fact can be explained by the larger size of the complexed anion.

The values of the formation constant *K_f_* of anion inclusion complexes with α-cyclodextrin as a function of temperature are presented on Figure 2 and Figure 3 and Table 2, Table 3, Table 4 and Table 5. As expected, we see that the *K_f_* values increase with the increasing molar mass of the studied salts and decrease with increasing temperature. The course of changes of the function *K_f_ = f(T)* and ln*K_f_* = *f*(1/*T*) is almost monotonic without any visible deviations (the value of the calculated R^2^ parameter for α-cyclodextrin is close to 1, which shows the linear dependence of ln*K_f_* = (1/*T*)).

There is no doubt that the *K_f_* values depend on the structure of the cyclodextrin molecules. In the case of α-cyclodextrin (degree of substitution 0.8) [25,26], it should be assumed that the studied anions are attached to the lower, smaller edge of the truncated cone (because this is the shape of the cyclodextrin molecules). Most likely, the interior of the cyclodextrin becomes less hydrophobic and, additionally, the anions of studied salts (which attach to the cyclodextrin) cannot completely fill its interior. It should be assumed that this may result in a much lower spontaneity of the complex formation process. The values of the complex formation constant are influenced by many factors, such as the size of the cyclodextrin gap as well as the presence of alkoxy and the number of hydroxyl groups in the studied anions. Thus, it should be assumed that the higher values of the formation constants (*K_f_*) of anion inclusion complexes with α-cyclodextrin may be due to a better match of the geometry of the anion molecules to the size of the cyclodextrin gap.

The conductometric measurements allowed for the determination of the thermodynamic functions of the complex formation (such as $\Delta G^{o}$-free enthalpy, $\Delta S^{o}$-entropy and $\Delta H^{o}$-enthalpy) of α-cyclodextrin with the tested salts of phenolic acids in water. These values are summarized in Table 2, Table 3, Table 4 and Table 5.

The temperature dependences of the complex formation constant were used to determine the free enthalpy of complex formation:(15)$$\Delta G^{o}\left(T\right)=-RT\ln K_{f}\left(T\right)$$

The above relationship can be presented as follows:(16)$$\Delta G^{o}\left(T\right)=A+BT+CT^{2}$$

The entropy values are presented as the first derivative of the free enthalpy as a function of temperature at constant pressure:(17)$$\Delta S^{o}=-{\left(\frac{\partial \Delta G^{o}}{\partial T}\right)}_{p}=-B-2CT$$

Enthalpy is calculated from the following relationship:(18)$$\Delta H^{o}=\Delta G^{o}+T\Delta S^{o}=A-CT^{2}$$

As can be seen from Table 2, Table 3, Table 4 and Table 5 and Figure 4, the free enthalpy ($\Delta G^{o}$) takes negative values for all tested salts over the entire range of analyzed temperatures.

Such a course of changes in this function indicates the spontaneity of the process of inclusion complex formation in the studied solutions. This effect is confirmed by the courses of changes in entropy ($\Delta S^{o}$) and enthalpy ($\Delta H^{o}$) as a function of temperature changes. The $\Delta H^{o}$ values decrease (become more and more negative) with increasing temperature, which indicates an increase in the exothermic effect of the formation of inclusion complexes (Figure 5). A similar nature of changes is observed in the case of the analysis of changes in the function $\Delta S^{o}$*= f (T)*, which is illustrated in Figure 6.

It should also be noted that all analyzed thermodynamic functions will assume increasingly negative values as the molar mass of the tested acids increases (cinnamic < coumaric < coffee < ferulic). Such a course of changes in the discussed thermodynamic functions can be explained by a different structure of these acids and consequently different energy which was needed to dehydrate their anions before they entered in the α-cyclodextrin gap. Most likely, the higher the molar mass of an anion, the lower the energy that is required for its dehydration.

## 5. Conclusions

The dependence of the complex formation constants allowed for the determination of thermodynamic functions in the studied temperature range. The results showed that the analyzed values of the standard free enthalpy of complex formation increases linearly with increasing temperature. The values of $\Delta H^{o}$ and $\Delta S^{o}$ functions of the complex formation process are negative over the entire temperature range. For the first time, the values of thermodynamic functions of complex formation were described due to the use of the conductometric method, which turns out to be universal in this type of operation. The values of *K_f_* as a function of temperature were estimated, and its influence on the discussed thermodynamic functions was shown. The process of formation of inclusion complexes is an exothermic reaction and is spontaneous. In the literature, there are papers concerning the determination of inclusion complexes of cyclodextrins [27,28,29], but they do not apply to the measurement method proposed in this paper and to selected research objects. This confirms the innovativeness of this work as well as the proposed innovative method of calculating the results.

## Figures and Tables

**Figure 1 molecules-27-04420-f001:**
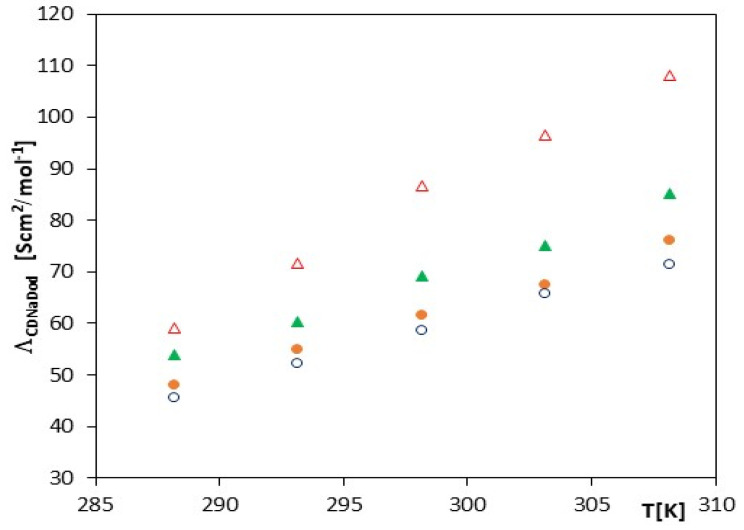
The plots of the dependence of the theoretical conductivity Λ*_CDNaDod_* (S∙cm^2^/mol^−1^) in the function *T* (K) for α-cyclodextrin with all studied salts: ∆-*trans*-cinnamic acid; ▲-*trans*-*p*-coumaric acid; ●-*trans*-caffeic acid; and ○-*trans*-ferulic acid.

**Figure 2 molecules-27-04420-f002:**
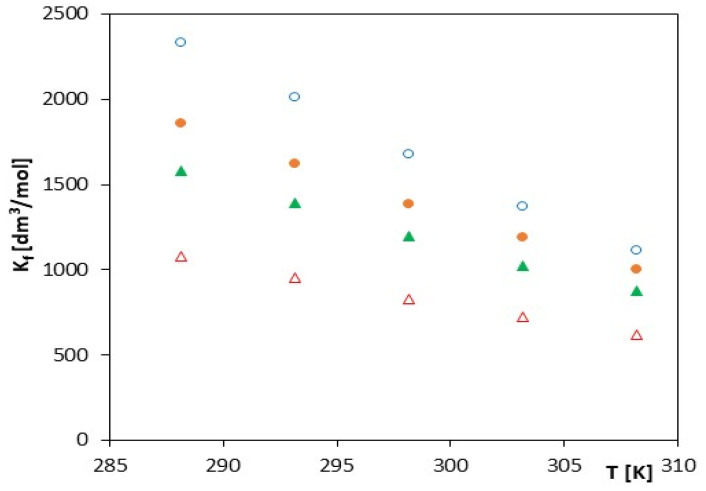
The plots of dependence of the *K_f_* (dm^3^/mol) formation constant in the *T* (K) function for α-cyclodextrin with all studied salts.: ∆-*trans*-cinnamic acid; ▲-*trans*-*p*-coumaric acid; ●-*trans*-caffeic acid; and ○-*trans*-ferulic acid.

**Figure 3 molecules-27-04420-f003:**
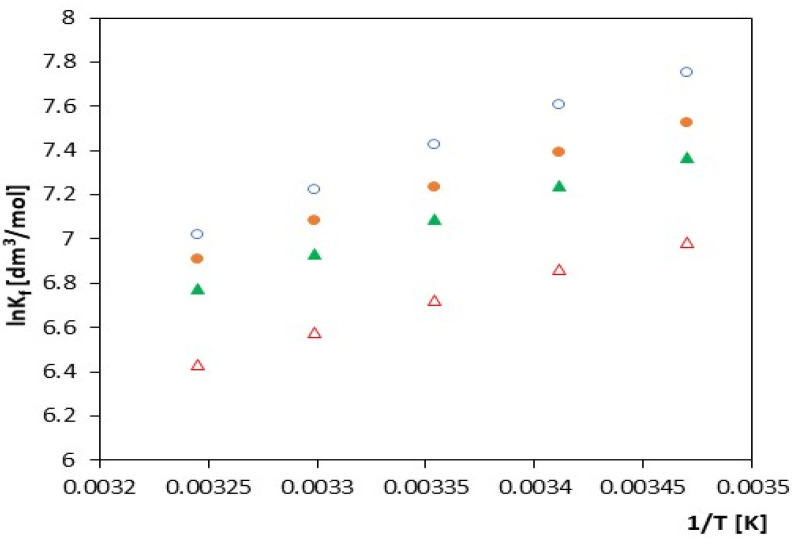
The plots of dependence of the lnK*_f_* (dm^3^/mol) formation constant in the *T* (K) function for α-cyclodextrin with all studied salts: ∆-*trans*-cinnamic acid; ▲-*trans-p*-coumaric acid; ●-*trans*-caffeic acid; and ○-*trans*-ferulic acid.

**Figure 4 molecules-27-04420-f004:**
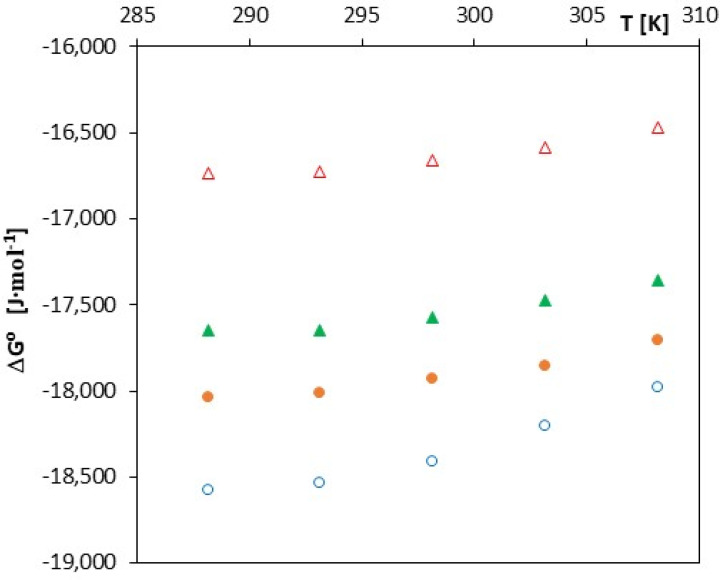
The plots of dependence of $\Delta G^{o}$ (J∙mol^−1^) in the function *T* (K) for α-cyclodextrin with all studied salts: ∆-*trans*-cinnamic acid; ▲-*trans*-*p*-coumaric acid; ●-*trans*-caffeic acid; and ○-*trans*-ferulic acid.

**Figure 5 molecules-27-04420-f005:**
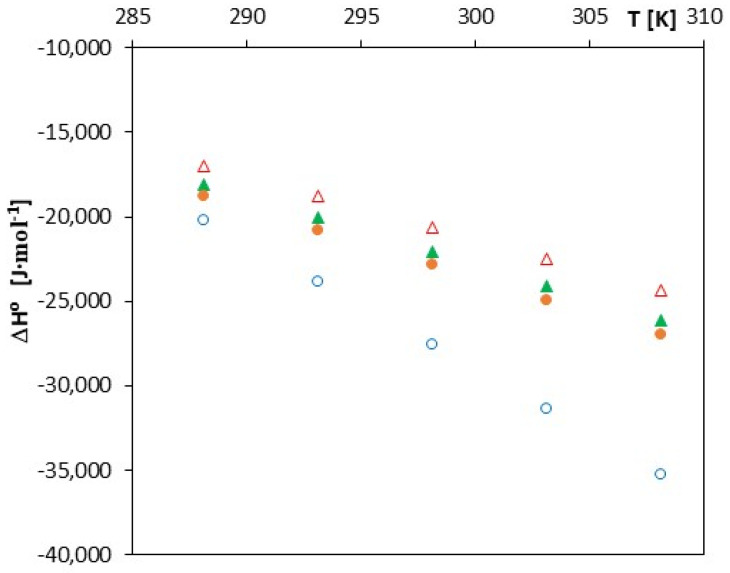
The plots of dependence of $\Delta H^{o}$ (J∙mol^−1^) in the function *T* (K) for α-cyclodextrin with all studied salts: ∆-*trans*-cinnamic acid; ▲-*trans-p*-coumaric acid; ●-*trans*-caffeic acid; and ○-*trans*-ferulic acid.

**Figure 6 molecules-27-04420-f006:**
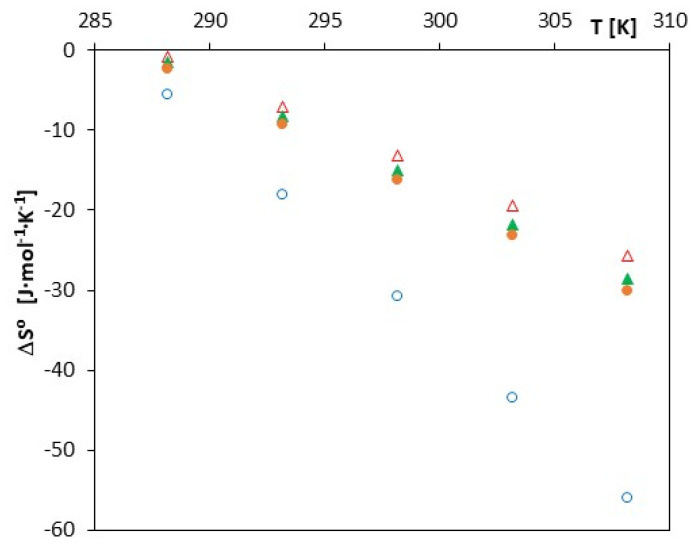
The plots of the dependence of $\Delta S^{o}$ (J∙mol^−1^∙K^−^^1^) in the function *T* (K) for α-cyclodextrin with all studied salts: ∆-*trans*-cinnamic acid; ▲-*trans-p*-coumaric acid; ●-*trans*-caffeic acid; and ○-*trans*-ferulic acid.

**Table 1 molecules-27-04420-t001:** Specification of chemical samples.

Chemical Name	ChemicalFormula	Chemical Formula	Source	CAS No	Mass FractionPurity
*trans*-ferulic acid	C_10_H_10_O_4_	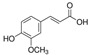	TCI *	537-98-4	≥0.998
*trans*-caffeic acid	C_9_H_8_O_4_	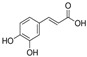	TCI *	331-39-5	≥0.998
*trans*-*p*-coumaric acid	C_9_H_8_O_3_	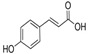	TCI *	501-98-4	≥0.998
*trans*-cinnamic acid	C_9_H_8_O_2_	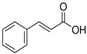	TCI *	140-10-3	≥0.998
α-Cyclodextrin	C_36_H_60_O_30_	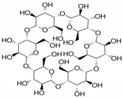	TCI *	10016-20-3	≥0.998
Sodium hydroxide micropills	NaOH		Avantor	1310-73-2	≥0.998

* TCI (Tokyo Chemical Industry).

**Table 2 molecules-27-04420-t002:** The value of formation constant *K_f_* (dm^3^/mol), theoretical conductivity Λ*_CDNaDod_* (S∙cm^2^/mol^−1^) and the values of thermodynamic functions $\Delta G^{o}$, $\Delta H^{o}$ and $\Delta S^{o}$ for α-cyclodextrin with the salt of *trans*-cinnamic acid.

*T* (K)	*K_f_* (dm^3^/mol)	ln*K_f_* (dm^3^/mol)	Λ*_CDNaDod_* (S∙cm^2^/mol^−1^)	$\Delta G^{o}$ (J∙mol^−1^)	$\Delta S^{o}$ (J∙mol^−1^∙K^−1^)	$\Delta H^{o}$ (J∙mol^−1^)	σ (Λ)
288.15	1080 ± 7	6.9847	59.23 ± 0.01	−16,733	−0.8350	−16,974	0.03
293.15	955 ± 3	6.8617	71.73 ± 0.01	−16,724	−7.0270	−18,784	0.02
298.15	830 ± 2	6.7214	86.62 ± 0.02	−16,661	−13.2190	−20,602	0.01
303.15	720 ± 2	6.5793	96.67 ± 0.01	−16,582	−19.4110	−22,467	0.01
308.15	620 ± 0.9	6.4297	108.24 ± 0.01	−16,473	−25.6030	−24,362	0.02

**Table 3 molecules-27-04420-t003:** The value of formation constant *K_f_* (dm^3^/mol), theoretical conductivity Λ*_CDNaDod_* (S∙cm^2^/mol^−1^) and the values of thermodynamic functions $\Delta G^{o}$, $\Delta H^{o}$ and $\Delta S^{o}$ for α-cyclodextrin with the salt of *trans*-*p*-coumaric acid.

*T* (K)	*K_f_* (dm^3^/mol)	ln*K_f_* (dm^3^/mol)	Λ*_CDNaDod_* (S∙cm^2^/mol^−1^)	$\Delta G^{o}$ (J∙mol^−1^)	$\Delta S^{o}$ (J∙mol^−1^∙K^−1^)	$\Delta H^{o}$ (J∙mol^−1^)	σ (Λ)
288.15	1583 ± 5	7.3671	54.04 ± 0.01	−17,649	−1.4670	−18,072	0.01
293.15	1395 ± 3	7.2406	60.44 ± 0.01	−17,647	−8.2319	−20,060	0.02
298.15	1197 ± 2	7.0876	69.12 ± 0.01	−17,569	−14.9970	−22,040	0.02
303.15	1025 ± 2	6.9324	75.12 ± 0.01	−17,472	−21.7620	−24,070	0.01
308.15	877 ± 0.8	6.7765	85.12 ± 0.01	−17,361	−28.5270	−26,152	0.02

**Table 4 molecules-27-04420-t004:** The value of formation constant *K_f_* (dm^3^/mol), theoretical conductivity Λ*_CDNaDod_* (S∙cm^2^/mol^−1^) and the values of thermodynamic functions $\Delta G^{o}$, $\Delta H^{o}$ and $\Delta S^{o}$ for α-cyclodextrin with the salt of *trans*-caffeic acid.

*T* (K)	*K_f_* (dm^3^/mol)	ln*K_f_* (dm^3^/mol)	Λ*_CDNaDod_* (S∙cm^2^/mol^−1^)	$\Delta G^{o}$ (J∙mol^−1^)	$\Delta S^{o}$ (J∙mol^−1^∙K^−1^)	$\Delta H^{o}$ (J∙mol^−1^)	σ (Λ)
288.15	1860 ± 8	7.5283	47.97 ± 0.02	−18,035	−2.3946	−18,725	0.02
293.15	1620 ± 6	7.3902	54.96 ± 0.01	−18,012	−9.3336	−20,748	0.02
298.15	1386 ± 2	7.2342	61.50 ±0.02	−17,932	−16.2726	−22,784	0.01
303.15	1192 ± 1	7.0834	67.54 ± 0.01	−17,853	−23.2116	−24,889	0.02
308.15	1004 ± 1	6.9117	76.13 ± 0.01	−17,708	−30.1506	−26,999	0.01

**Table 5 molecules-27-04420-t005:** The value of formation constant *K_f_* (dm^3^/mol), theoretical conductivity Λ*_CDNaDod_* (S∙cm^2^/mol^−1^) and the values of thermodynamic functions $\Delta G^{o}$, $\Delta H^{o}$ and $\Delta S^{o}$ for α-cyclodextrin with the salt of *trans*-ferulic acid.

*T* (K)	*K_f_* (dm^3^/mol)	ln*K_f_* (dm^3^/mol)	Λ*_CDNaDod_* (S∙cm^2^/mol^−1^)	$\Delta G^{o}$ (J∙mol^−1^)	$\Delta S^{o}$ (J∙mol^−1^∙K^−1^)	$\Delta H^{o}$ (J∙mol^−1^)	σ(Λ)
288.15	2335 ± 4	7.7558	45.68 ± 0.01	−18,580	−5.5348	−20,175	0.02
293.15	2010 ± 3	7.6059	52.23 ± 0.02	−18,537	−18.1488	−23,858	0.01
298.15	1680 ± 1	7.4265	58.67 ± 0.01	−18,409	−30.7628	−27,581	0.01
303.15	1370 ± 1	7.2226	65.68 ± 0.01	−18,204	−43.3768	−31,353	0.03
308.15	1116 ± 1	7.0175	71.52 ± 0.01	−17,979	−55.9908	−35,232	0.02

## Data Availability

Not applicable.

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
