# Peer review of "Studies of the Formation of Inclusion Complexes Derivatives of Cinnamon Acid with α-Cyclodextrin in a Wide Range of Temperatures Using Conductometric Methods"

_molecules, 2022, doi:10.3390/molecules27144420_

Round 1

Reviewer 1 Report

Dear Editor,

There are almost sufficient works have been done in this paper with the title "Studies of formation of inclusion complexes derivatives of cin-2 namon acid with -cyclodextrin in a wide range of tempera-3 tures using conductometric methods." to achieve a useful product that in my view it is good to have accepted in the journal of molecules. However, there are some parts that needed to be corrected which are sorted below:

1. The abstract is incomplete, it should contain all of the detailed information about the prominent achievements of the experiments including numeric results.

2. In the introduction, goals should be mentioned in the latest paragraph but the authors report on the work process. It needs to be changed.

3. In the experimental section, it is better to depict any test with a flowchart or simple schematics.

4.2. Results convert to 3. Results and 3. discussion converts to 4. conclusion.

5. In the conclusion, the writer should bring their numeric results with their discussions altogether.

6. Unfortunately, I could not find any novelty in this work. Please elucidate it.

Author Response

Answer for the Reviewer 1

           I would like to thank you for the constructive comments and corrections of the manuscript to enhance its scientific values. In the revised version of my paper I included all comments and suggestions. I hope that the responses to the comments/suggestions presented below are  sufficient and satisfactory. In the revised version of the work, all the changes are marked in yellow. Our responses to your comments are below.

  1. As suggested by the Reviewer, the following changes were made in the Abstract (a sentence was added at the end):

“For the first time in literature, the values of thermodynamic functions ΔGo, ΔHo and ΔSo of complex formation of studied acids with α-cyclodextrin were described due to the use of the conductometric method as a function of temperature changes. The obtained results show that the process of formation of inclusion complexes is an exothermic reaction and is spontaneous.”

  1. In accordance with the Reviewer's suggestions, the Introduction section has been redrafted. At the beginning of this chapter was added:

“The main objectives of the presented study were to determine the values of the constants formation (Kf) of inclusion complex  of cinnamon acid and its selected derivatives: trans-4-hydroxycinnamic acid, trans-3,4-dihydroxycinnamic acid, trans-4-hydroxy-3-methoxycinnamic acid, and trans-3-phenylacrylic acid with a-cyclodextrin based on conductometric measurements and analysis of the impact of these acids structure on the discussed values Kf.

4.2. Chapters were renumbered in the article: 3. Results, 4. Discussion and 5. Conclusion

  1. All obtained in this paper results are summarized in Tables 2-5 and the Table in Supplementary Material.

  1. The novelty of this work is that for the first time in the literature, the equations modified by us were presented, allowing the use of conductometric tests to calculate the constants of formation of cyclodextrin inclusion complexes.

The revised version of the article includes corrections to the English language recommended by a native English speaker.

Reviewer 2 Report

Comments. Authors report a study on the determination of thermodynamic parameters concerning inclusion complexes between cinnamon acid and α-cyclodextrin. Study was conducted using the limiting molar conductivity, obtained from conductivity measurements, for the determination of the complexation constants (Kf) and of the thermodynamic functions ??, ?H and ?S of the complexes.

I think that the procedure followed in the manuscript is interesting and well described.

Manuscript can be accepted for the publication on Molecules and I report just some minor comments

1.      A figure with the structures of the cinnamic acids under study could be added to help the reader

2.      Pag. 3, lines 109-110 and 132-133 the sentences “All solutions were prepared….” report the same information. One of the two sentences must be deleted

3.      Pag. 4, line 149, sentence “In the studied solutions, we observe the formation of inclusion complexes…. ”  I think it is not the correct term as there is no experimental evidence but the formation of the complexes is obtained from data processing

4.      Pag 4, line 164: “….. calculated from Equation (5), ……”. I think is eq (3).

Author Response

       I would like to thank you for the constructive comments and corrections of the manuscript to enhance its scientific values. In the revised version of my paper I included all comments and suggestions. I hope that the responses to the comments/suggestions presented below are  sufficient and satisfactory. In the revised version of the work, all the changes are marked in yellow. Our responses to your comments are below.

1. Thank you for your suggestion – The structures of the tested acids are added in Table 1

2. Removed sentence: “All solutions were prepared gravimetrically using a balance analytical apparatus (Sartorius RC 210D) with an accuracy of ± 1⋅10-5 g.” from lines 132-133 .

3.  Page 4 line 149 – should be:

“In the studied solutions, we observe most probably the formation of inclusion complexes with the 1: 1 stoichiometry between the cyclodextrin (CD) and the anion of the tested phenolic acids (Dod-).”

4.  Thank you for your attention, it is my mistake. Should be: Equation 3

The revised version of the article includes corrections to the English language recommended by a native English speaker.

Reviewer 3 Report

Please provide a graphical abstract for the article, to increase appeal and accessibility.

Please insert the chemical structures of the compounds discussed (those in table 1).

The impact and relevance of the work performed must be better underlined, perhaps by elaborating more in the discussion section.

References are scarce and rather old. Please provide newer references. A small number of new references suggest that the object of study might not be relevant in the present.

1.       Line 34 “nature of the [3, 4].” There seems to be something missing.

2.       Line 68 “They are widespread throughout the country in the plant kingdom” please revise.

3.       Line 71 “benzoic acids, phenylacetic acids, and 71 cinnamon acids” should be rephrased.

Author Response

 I would like to thank you for the constructive comments and corrections of the manuscript to enhance its scientific values. In the revised version of my paper I included all comments and suggestions. I hope that the responses to the comments/suggestions presented below are  sufficient and satisfactory. In the revised version of the work, all the changes are marked in yellow. Our responses to your comments are below.

  • In the revised version of our article, Graphical Abstract has been corrected.
  • The structures of the tested acids are added in Table 1
    • As suggested by the Reviewer: Abstract and Introduction have been partially modified.
    • In the literature from recent years, there are no data on the conductometric, calorimetric or potentiometric studies used to calculate the constants of formation of inclusion complexes cyclodextrins with presented in this paper acids. Therefore, it is not possible to compare the results obtained by us with the research of other authors.
    • Taking into account the suggestions of the Reviewer was added to Conclusion:

    “In the literature, there are papers concerning the determination of inclusion complexes of cyclodextrins [27-29], but they do not apply to the measurement method proposed in this paper and to selected research objects. This confirms the innovativeness of this work as well as the proposed innovative method of calculating the results.”

    and in the References position 27-29.

    1. Line 34 “nature of the [3, 4].

    Thank you for your attention, it is my mistake. Should be:

     “The hydrogen atoms located in the C3 and C5 carbon atoms and the oxygen atom  the glucopyranose ring are directed toward the inside of the molecule, ensuring the hydrophobic nature of the torus [3, 4].”

    .       2. Line 68 “They are widespread throughout the country in the plant kingdom” please revise.

                Should be:

            “Phenolic acids have an antioxidant potential that is greatest for the cinnamic acid derivatives. They are widespread throughout  in the word of plants, and although they are characterized by a variety of chemical structures and pharmacological properties, they contain the same functional groups (hydroxyl and carboxyl groups) [10].”

    1. Line 71 “benzoic acids, phenylacetic acids, and cinnamon acids” should be rephrased.

    Thank you for your suggestion. Should be:

    “The most common derivatives of cinnamic acid are its hydroxyl derivatives, i.e. caffeic acid, ferulic acid and p-coumaric acid. Phenolic acids are commonly present in food and dietary supplements as bioactive ingredients.”

    The revised version of the article includes corrections to the English language recommended by a native English speaker.

Reviewer 4 Report

The quality of the figures should be well-improved.

The authors should provide sufficient data and literature to support methods or results of their works.

Author Response

Answer for the Reviewer 4

Thank you so much for your review. Our responses to your comments are below.

  • The values of the standard uncertainties, combined expanded uncertainty and standard deviation for the determined values of molar conductivity are presented in Tables 2-5 and in the Table in Supplementary Materials, what makes that the results presented in Tables and Figures are more reliable.

  • In the literature from recent years, there are no data on the conductometric, calorimetric or potentiometric studies used to calculate the constants of formation of inclusion complexes cyclodextrins with presented in this paper acids. Therefore, it is not possible to compare the results obtained by us with the research of other authors.

Taking into account the suggestions of the Reviewer was added to Conclusion:

“In the literature, there are papers concerning the determination of inclusion complexes of cyclodextrins [27-29], but they do not apply to the measurement method proposed in this paper and to selected research objects. This confirms the innovativeness of this work as well as the proposed innovative method of calculating the results.”

and in the References position 27-29.

  • The quality of the Figures 1-6 have  been improved.

The revised version of the article includes corrections to the English language recommended by a native English speaker.

Round 2

Reviewer 1 Report

Dear Editor,

Some suggestions have been answered and the article can be accepted.

Author Response

Answer for the Reviewer 1

           I would like to thank you for the constructive comments and corrections of the manuscript to enhance its scientific values.

Reviewer 3 Report

The authors have provided the requested alterations. 

Author Response

Answer for the Reviewer 3

           I would like to thank you for the constructive comments and corrections of the manuscript to enhance its scientific values.

Reviewer 4 Report

1. What are the solubility of the research materials? It is suggested to provide a more detailed solution preparation method required in this study.

2. What is the error of the experimental data? It is recommended to add error bars to the Figures.

Author Response

Answer for the Reviewer 4

Thank you so much for your review. Our responses to your comments are below.

  1. The solubility of the tested sodium salts in water in the studied concentration range was very good and thus did not pose any experimental problems in conductivity measurements (as suggested by the Reviewer, this sentence was added to the article). The exact procedure for the preparation of the tested salts of phenolic acids is given in the articles cited in the reviewed article in positions [12, 16, 19, 20].

  1. The values of the experimental error of the determined conductivity values as well as the error for the values of the complex formation constants Kf are presented in the Tables included in the main article and in Supplementary Materials. Placing the error bars suggested by the Reviewer on the charts makes them completely unreadable, as presented in the chart below.
